# Algorithms and Hardness for Estimating Statistical Similarity

## Abstract

We introduce and study the computational problem of determining statistical similarity between probability distributions. For distributions $P$ and $Q$ over a finite sample space, their statistical similarity is defined as $S_{\text{stat}}(P,Q) := \sum_x \min(P(x), Q(x))$. Despite its fundamental nature as a measure of similarity between distributions, capturing essential concepts such as Bayes error in prediction and hypothesis testing, this computational problem has not been previously explored. Recent work on computing statistical distance has established that, somewhat surprisingly, even for the simple class of product distributions, exactly computing statistical similarity is #P-hard. This motivates the question of designing approximation algorithms for statistical similarity. Our first contribution is a Fully Polynomial-Time deterministic Approximation Scheme (FPTAS) for estimating statistical similarity between two product distributions. Furthermore, we also establish a complementary hardness result. In particular, we show that it is NP-hard to estimate statistical similarity when $P$ and $Q$ are Bayes net distributions of in-degree 2.

## 1 Introduction

Given two distributions $P$ and $Q$ over a finite sample space $D$, their statistical similarity, denoted $S_{\text{stat}}(P,Q)$, is defined as

$$S_{\text{stat}}(P,Q) := \sum_{x \in D} \min(P(x), Q(x)). \tag{1}$$

Statistical similarity serves as a fundamental measure in machine learning and statistical inference. We defer a detailed discussion of motivating applications to Section 1.1.

When the sample space is small, computing $S_{\text{stat}}$ is trivial. However, for high-dimensional distributions, this computation presents significant challenges. Surprisingly, recent work (Bhattacharyya et al., 2023) has established that computing $S_{\text{stat}}$ is #P-hard even for the simple class of product distributions. This hardness result is striking given that product distributions represent one of the most basic high-dimensional distribution classes, where each dimension is independent of other dimensions. The hardness of this elementary case raises fundamental questions about the computational nature of statistical similarity: Can we develop efficient approximation algorithms for classes of distributions of interest? In general, what is the boundary between tractable and intractable similarity computation?

The primary contribution of this work is to initiate a principled investigation of the computational aspects of statistical similarity, identifying both tractable and intractable scenarios. Our first contribution is a Fully Polynomial-Time deterministic Approximation Scheme (FPTAS) for estimating $S_{\text{stat}}$ between product distributions. To complement this algorithmic result, we establish sharp computational boundaries by proving that approximating $S_{\text{stat}}$ becomes NP-hard even for slightly more general distributions. Specifically, we show that the problem is NP-hard to approximate for Bayes net distributions with in-degree 2. Note that we work in a computational setting, where the algorithms have access to a succinct description of distributions.

### 1.1 Motivating Applications

Statistical similarity plays a central role across multiple domains in machine learning and statistics. We examine three key applications of statistical similarity: Its connection to Bayes error in prediction

problems, its role in characterizing optimal decision rules in hypothesis testing, and its interpretation through coupling theory. These applications demonstrate the significance of $S_{\text{stat}}$.

Statistical similarity arises naturally in the analysis of prediction problems through the notion of *Bayes error*. Consider a binary prediction problem defined by a distribution $P$ over $X \times \{0, 1\}$, where $X$ is a finite feature space. When a classifier $g : X \to \{0, 1\}$ attempts to predict the label, it incurs a 0-1 prediction error measured as $\mathbf{Pr}_{(x,y) \sim P}[g(x) \neq y]$. The *Bayes optimal classifier*, which outputs 1 if and only if $P(1|x) > P(0|x)$, achieves the minimum possible error $R^*$, known as the Bayes error. This error represents a fundamental lower bound that no classifier can surpass. The connection to statistical similarity manifests through a precise mathematical relationship: For any prediction problem, the Bayes error exactly equals the statistical similarity between its scaled likelihood distributions. Specifically, if we denote the prior probabilities $P(0)$ and $P(1)$ by $\alpha_0$ and $\alpha_1$ respectively, then $R^* = S_{\text{stat}}(\alpha_0 P(X|0), \alpha_1 P(X|1))$ (a proof is given in Appendix A), where $\alpha_i P(X|i)$ represents the sub-distribution obtained by scaling $P(i|X)$ with $\alpha_i$.

The relationship between statistical similarity and optimal decision-making extends beyond prediction problems to the domain of hypothesis testing (Lehmann & Romano, 2008; Nielsen, 2014). This setting is particularly relevant to our computational focus, as it deals with known distributions representing null and alternate hypotheses. A recent result (Kontorovich & Avital, 2024) establishes how statistical similarity between *product distributions* determines the optimal error in hypothesis testing (Parisi et al., 2014; Berend & Kontorovich, 2015). To illustrate this connection, consider a hypothesis testing game where a random bit $Y \in \{0, 1\}$ is drawn with bias $p_1$ (letting $p_0 = 1 - p_1$), followed by an i.i.d. sequence $X_1, \ldots, X_n$ where each $X_i \in \{0, 1\}$ satisfies $\mathbf{Pr}[X_i = 1|Y = 1] = \psi_i$ and $\mathbf{Pr}[X_i = 1|Y = 0] = \eta_i$ for parameters $\psi, \eta \in (0, 1)^n$. The optimal decision rule $f^{\text{OPT}} : \{0, 1\}^n \to \{0, 1\}$ that minimizes $\mathbf{Pr}[f^{\text{OPT}}(X) \neq Y]$ achieves an error rate of $S_{\text{stat}}(p_1 \text{Bern}(\psi), p_0 \text{Bern}(\eta))$, where $\text{Bern}(\psi)$ denotes the product distribution of individual $\text{Bern}(\psi_i)$ distributions.

These theoretical connections have significant practical implications. Since Bayes error represents the theoretically optimal performance limit, statistical similarity serves as a benchmark for the evaluation of machine learning models. This capability has spurred extensive research in estimating Bayes error and statistical similarity (Fukunaga & Hostetler, 1975; Devijver, 1985; Noshad et al., 2019; Theisen et al., 2021; Ishida et al., 2023).

Statistical similarity can be interpreted through coupling theory. For distributions $P$ and $Q$, a coupling is a distribution $(X, Y)$ where $X \sim P$ and $Y \sim Q$. It is known that $S_{\text{stat}}(P, Q)$ equals the maximum over all couplings $(X, Y)$, $\mathbf{Pr}(X = Y)$. Coupling theory, introduced by Doeblin (1938), has led to important results in computer science and mathematics (Lindvall, 2002; Levin et al., 2006; Meyn & Tweedie, 2012). Finally, statistical similarity admits a characterization in the form of statistical distance (also known as total variation distance) $d_{\text{TV}}$, defined as $d_{\text{TV}}(P, Q) := \max_{S \subseteq D}(P(S) - Q(S)) = \frac{1}{2} \sum_{x \in D} |P(x) - Q(x)|$. The identity $S_{\text{stat}}(P, Q) = 1 - d_{\text{TV}}(P, Q)$, known as Scheffé's identity, establishes a duality (see Appendix B).

## 1.2 PAPER ORGANIZATION

We present some necessary background material in Section 2. We then present a survey of related work in Section 3. Section 4 describes our primary contributions. Section 5 is dedicated to our algorithmic result. The proof of NP-hardness of estimating the statistical similarity between in-degree 2 Bayes net distributions is provided in Section 6. Section 7 gives some concluding remarks. Appendix A discusses the connections between Bayes error and statistical similarity. Similarly, Appendix B elaborates on the connection between TV distance and statistical similarity. Appendix C contains the proof of Claim 11, used in the proof of Theorem 6.

## 2 PRELIMINARIES

We use $[n]$ to denote the set $\{1, \ldots, n\}$. We will use $\log$ to denote $\log_2$. The following notion of a deterministic approximation algorithm is important in this work.

**Definition 1.** A function $f : \{0, 1\}^* \to \mathbb{R}$ admits a *fully polynomial-time approximation scheme (FPTAS)* if there is a *deterministic* algorithm $\mathcal{A}$ such that for every input $x$ (of length $n$) and $\varepsilon > 0$, the algorithm $\mathcal{A}$ outputs a $(1 + \varepsilon)$-multiplicative approximation to $f(x)$, i.e., a value that lies in the interval $[f(x)/(1 + \varepsilon), (1 + \varepsilon)f(x)]$. The running time of $\mathcal{A}$ is $\text{poly}(n, 1/\varepsilon)$.

**Definition 2.** Given two distributions $P$ and $Q$ over a finite sample space $D$, the statistical similarity between $P$ and $Q$ is $S_{\text{stat}}(P, Q) := \sum_{x \in D} \min(P(x), Q(x))$.

A product distribution $P$ over $[\ell]^n$ can be described by $n$ functions $p_1, \ldots, p_n$ such that $p_i(x) \in [0, 1]$ is the probability that the $i$-th coordinate equals $x \in [\ell]$. For any $y \in [\ell]^n$, the probability of $y$ with respect to $P$ is given by $P(y) = \prod_{i=1}^n p_i(y_i)$.

We require the following.

**Proposition 3** (See also Lemma 3 in Kontorovich (2012)). *For product distributions $P = P_1 \otimes \cdots \otimes P_n$ and $Q = Q_1 \otimes \cdots \otimes Q_n$, it is the case that $S_{\text{stat}}(P, Q) \geq \prod_{i=1}^n S_{\text{stat}}(P_i, Q_i)$.*

*Proof.* We will utilize a coupling argument. Let $\mathcal{O} = (X, Y)$ be an *optimal* coupling between $P$ and $Q$, i.e., $\mathbf{Pr}_{\mathcal{O}}[X = Y] \geq \mathbf{Pr}_{\mathcal{C}}[X = Y]$ for any coupling $\mathcal{C}$. Thus, $\mathbf{Pr}_{\mathcal{O}}[X = Y] = S_{\text{stat}}(P, Q)$ (as mentioned in Section 1.1). For $1 \leq i \leq n$, let $\mathcal{O}_i = (X_i, Y_i)$ be an optimal coupling between $P_i$ and $Q_i$. That is, $X_i \sim P_i, Y_i \sim Q_i$ and $\mathbf{Pr}_{\mathcal{O}_i}[X_i = Y_i] = S_{\text{stat}}(P_i, Q_i)$. Let $O'$ be the coupling given by the product of $O_i$'s. Then

$$S_{\text{stat}}(P, Q) = \mathbf{Pr}_{\mathcal{O}}[X = Y] \geq \mathbf{Pr}_{\mathcal{O}'}[X = Y] = \prod_{i=1}^n \mathbf{Pr}_{\mathcal{O}_i}[X_i = Y_i] = \prod_{i=1}^n S_{\text{stat}}(P_i, Q_i). \qquad \square$$

**Proposition 4.** *Let $P = P_1 \otimes \cdots \otimes P_n$ and $Q = Q_1 \otimes \cdots \otimes Q_n$ be product distributions over $[\ell]^n$. Let $\tau_P$ be a lower bound on $P_i(x)$ for any $i$ and $x \in [\ell]$ whereby $P_i(x)$ is nonzero. Similarly define $\tau_Q$ and let $\tau := \min(\tau_P, \tau_Q)$. Then if $\prod_{i=1}^n S_{\text{stat}}(P_i, Q_i) > 0$, then it is the case that $\prod_{i=1}^n S_{\text{stat}}(P_i, Q_i) \geq \tau^n$.*

*Proof.* It would suffice to show that $S_{\text{stat}}(P_i, Q_i) \geq \tau$, since the $P_i$'s and the $Q_i$'s are independent. Since $\prod_{i=1}^n S_{\text{stat}}(P_i, Q_i) > 0$, for all $i$ there must be some $x$ such that $P_i(x), Q_i(x) > 0$ and $\min(P_i(x), Q_i(x)) > \tau$. Therefore,

$$S_{\text{stat}}(P_i, Q_i) = \sum_{x \in [\ell]} \min(P_i(x), Q_i(x)) \geq \tau. \qquad \square$$

**Definition 5** (*J*-Sparsification (Feng et al., 2024)). Let $S$ be a discrete random variable that takes values between 0 and $B$ and $J$ be a collection of intervals $I_0, \ldots, I_m$ that partition $[0, B]$. We define the *J-sparsification* of $S$, denoted by $\widetilde{S}$, as follows: Given an interval $I_j$, let $S_j$ be the conditional expectation of $S$ conditioned on $S$ lying in the interval $I_j$. That is, suppose that $S$ takes values $r_1, \ldots, r_k$ in the interval $I_j$. Then $S_j = \sum_{i=1}^k \mathbf{Pr}[S = r_i] r_i / \sum_{i=1}^k \mathbf{Pr}[S = r_i]$. Now the sparsified random variable $\widetilde{S}$ takes value $S_j$ with probability $\mathbf{Pr}[S \in I_j] = \sum_{i=1}^k \mathbf{Pr}[S = r_i]$.

# 3 RELATED WORK

Estimating the Bayes error has been a topic of continued interest in the machine learning community Fukunaga & Hostetler (1975); Devijver (1985); Noshad et al. (2019); Theisen et al. (2021); Ishida et al. (2023). These works focus on the setting where distributions are only accessible through samples rather than explicitly specified, leading to techniques distinct from those needed in our setting of explicitly represented distributions.

The computational complexity of statistical similarity was established through Scheffé's identity $(S_{\text{stat}}(P, Q) + d_{\text{TV}}(P, Q) = 1)$ and the result of Bhattacharyya et al. (2023), where it is shown that the exact computation of $d_{\text{TV}}$ (and thus $S_{\text{stat}}$) is #P-hard even for product distributions. This hardness naturally leads to the study of approximation algorithms, with multiplicative approximation being stronger than additive approximation for measures bounded in $[0, 1]$.

For distributions samplable by Boolean circuits, additive approximation of statistical similarity is complete for SZK (Statistical Zero Knowledge) (Sahai & Vadhan, 2003), while the problem becomes tractable for distributions that are both samplable and have efficiently computable point probabilities (Bhattacharyya et al., 2020).

While recent work has made significant progress on multiplicative approximation of statistical distance (also known as total variation distance), including an FPRAS (Feng et al., 2023) and an FPTAS (Feng

et al., 2024) for product distributions, these results do not directly translate to statistical similarity. This is because multiplicative approximation of statistical distance does not yield multiplicative approximation of its complement (statistical similarity), necessitating new algorithmic techniques for statistical similarity. Similarly, the NP-hardness result for multiplicatively approximating statistical distance between Bayes nets (Bhattacharyya et al., 2023) does not immediately imply hardness for statistical similarity.

It is perhaps worth remarking that technical barrier in translating multiplicative approximation of statistical distance to statistical similarity is rather fundamental, i.e., it is not possible in general to use an efficient multiplicative approximation algorithm for a function $f$ in order to design an efficient multiplicative approximation algorithm for $1 - f$. In particular, even if there is an efficient multiplicative approximation algorithm $f$, approximating $1 - f$ could be NP-hard. For instance, let $f$ be a function that takes as input a Boolean DNF formula $\phi$ and outputs the probability that a random assignment satisfies $\phi$. It is known that there is a randomized multiplicative approximation algorithm for estimating $f$ Karp et al. (1989). However, a multiplicative approximation algorithm for estimating $1 - f$ implies that all NP-complete problems have efficient randomized algorithms (RP = NP). This is because the complement of a DNF formula is a CNF formula, and there is no efficient randomized multiplicative approximation for estimating the acceptance probability of CNF formulas unless RP = NP.

The connection between statistical similarity and hypothesis testing has been explored in several works. While Kontorovich & Avital (2024) provides analytical bounds on statistical similarity for product distributions in the context of hypothesis testing, these bounds do not yield multiplicative approximation algorithms.

## 4 OUR RESULTS

Our first contribution is the design of a deterministic polynomial-time approximation scheme to estimate the statistical similarity between product distributions.

**Theorem 6.** *There is an FPTAS for estimating $S_{\text{stat}}(P, Q)$ whereby $P$ and $Q$ are product distributions succinctly represented by their parameters.*

Theorem 6 is proved by adapting the ideas of Feng et al. (2024). We define a random variable $R = P \| Q$ which is the ratio of $P$ and $Q$ and then partition its range into a sequence of intervals. Every one of these intervals is subsequently "sparsified," in the sense that we only take into account the average value of $R$ over this interval. This allows us to efficiently estimate statistical similarity, as we show in the proof.

A natural question is whether Theorem 6 can be extended to more general distributions such as Bayes net distributions. Our second result is a hardness result.

**Theorem 7.** *Given two probability distributions $P$ and $Q$ that are defined by Bayes nets of in-degree two, it is NP-complete to decide whether $S_{\text{stat}}(P, Q) \neq 0$ or not. Hence the problem of multiplicatively estimating $S_{\text{stat}}$ is NP-hard.*

Theorem 7 is proved by adapting the proof of hardness of approximating TV distance between Bayes net distributions presented in Bhattacharyya et al. (2023).

## 5 ESTIMATING STATISTICAL SIMILARITY

We prove Theorem 6. Let $P, Q$ be distributions and $D$ be the common domain of $P, Q$ and let $R$ be a ratio random variable that takes the value $P(x) / Q(x) \geq 0$ with probability $Q(x)$. We can assume $Q(x) > 0$, as $R$ only takes value when $x$ is such that $Q(x) > 0$. We denote this by writing $R := P \| Q$. For ratios $R_1 = P_1 \| Q_1, R_2 = P_2 \| Q_2$ we define their independent product $R_1 \cdot R_2$ as a random variable that takes the value $(P_1(x) / Q_1(x)) (P_2(y) / Q_2(y))$ with probability $Q_1(x) Q_2(y)$.

Moreover, we overload notation and for a ratio random variable $R = P \| Q$, we let $S_{\text{stat}}(R)$ denote the functional $\mathbf{E}[\min(R, 1)]$. Then

$$S_{\text{stat}}(R) = \mathbf{E}[\min(R, 1)]$$

$$= \mathop{\mathbf{E}}_{x \sim Q}[\min(P(x)/Q(x), 1)]$$

$$= \sum_x \min(P(x)/Q(x), 1) Q(x) = \sum_x \min(P(x), Q(x)) = S_{\text{stat}}(P, Q).$$

**Setting and Algorithm Definition.** We denote by $\varepsilon$ the desired accuracy parameter. Let $\tau_P$ be a lower bound on $P_i(x)$ for any $i$ and $x \in [\ell]$ whereby $P_i(x)$ is nonzero. Similarly define $\tau_Q$ and let $\tau := \min(\tau_P, \tau_Q)$. Let also $B := 1/\tau^n$, $\delta := (1 + \varepsilon/2)^{1/n} - 1$, and $\gamma := \tau^{2n}(\varepsilon/2)\delta/(n(1+\delta)^n)$.

We require the following.

**Proposition 8.** *It is the case that* $\mathbf{Pr}[0 \le R \le B] = 1$.

*Proof.* By definition, $R \ge 0$. Since $R$ takes the value $P(x)/Q(x)$ with probability $Q(x)$, we get that $R$ is at most $\max_x(P(x)/Q(x)) \le 1/\tau^n = B$ with probability 1. $\square$

We now define a set of intervals

$$J := \left\{\{0\}, I_0 := (0, \gamma], I_1 := (\gamma, \gamma(1+\delta)], \dots, I_m := (\gamma(1+\delta)^{m-1}, \gamma(1+\delta)^m = B]\right\},$$

whereby $m := (\log(B/\gamma))/\log(1+\delta)$. Define $R_1, \dots, R_n$ to be the ratios for each coordinates, that is, $R_i := P_i \| Q_i$. We define a set of random variables $Y_i$ for $1 \le i \le n$. Define $Y_1 = R_1$ and $Y_{i+1} = \widetilde{Y}_i \cdot R_{i+1}$ where $\widetilde{Y}_i$ is the $J$-sparsification of $Y_i$. Also, for convenience, set $Z_i := R_{i+1} \cdot \dots \cdot R_n$ and $Z_n = 1$. The output of our algorithm is $S_{\text{stat}}(\widetilde{Y}_n \cdot Z_n) = S_{\text{stat}}(\widetilde{Y}_n)$. See Algorithm 1.

---

**Algorithm 1** The pseudocode of our algorithm.

---

**Require:** Product distributions $P, Q$ through their marginal distributions $P_1, \dots, P_n, Q_1, \dots, Q_n$, each over $[\ell]$, and an accuracy error parameter $\varepsilon$.

**Ensure:** The output $S_{\text{stat}}\left(\widetilde{Y}_n\right)$ is an $\varepsilon$-approximation of $S_{\text{stat}}(P, Q)$.

1: {We can compute $n$ by parsing the input.}
2: **for** $i \leftarrow 1, \dots, n$ **do**
3:    $R_i \leftarrow P_i \| Q_i$
4:    {Computing $S_{\text{stat}}(R_i)$ takes time $O(\ell)$.}
5:    **if** $S_{\text{stat}}(R_i) = 0$ **then**
6:       **return** 0
7:    **end if**
8: **end for**
9: $\delta \leftarrow (1 + \varepsilon/2)^{1/n} - 1$
10: $\tau_P \leftarrow \min\{P_i(x) \mid i \in [n], x \in [\ell], P_i(x) > 0\}$          {This step takes time $O(n\ell)$.}
11: $\tau_Q \leftarrow \min\{Q_i(x) \mid i \in [n], x \in [\ell], Q_i(x) > 0\}$          {This step takes time $O(n\ell)$.}
12: $\tau \leftarrow \min(\tau_P, \tau_Q)$
13: $\gamma \leftarrow \tau^{2n}(\varepsilon/2)\delta/(n(1+\delta)^n)$
14: $J \leftarrow \{\{0\}, \{(0, \gamma]\}\}$
15: **for** $i \leftarrow 1, \dots, m$ **do**
16:    $J \leftarrow J \cup \left\{\left(\gamma(1+\delta)^{i-1}, \gamma(1+\delta)^i\right]\right\}$
17: **end for**
18: $Y_1 \leftarrow R_1$
19: **for** $i \leftarrow 1, \dots, n$ **do**
20:    $\widetilde{Y}_i \leftarrow J$-sparsification of $Y_i$          {This step takes time $O(m\ell)$.}
21:    **if** $i < n$ **then**
22:       $Y_{i+1} \leftarrow \widetilde{Y}_i \cdot R_{i+1}$          {This step takes time $O(m\ell)$.}
23:    **end if**
24: **end for**
25: **return** $S_{\text{stat}}\left(\widetilde{Y}_n\right)$          {Computing $S_{\text{stat}}\left(\widetilde{Y}_n\right)$ takes time $O(m)$.}

---

**Running Time.** Note that $S_{\text{stat}}(R_i) = S_{\text{stat}}(P_i, Q_i)$ can be computed in time $O(\ell)$ by following the equality $S_{\text{stat}}(R_i) = \mathbf{E}[\min(R_i, 1)]$ and utilizing the fact that $R_i$ may assume $\ell$ values. Moreover, the sparsification step can be computed in time $O(m\ell)$. This is because, firstly, in Line 22 we generate $m\ell$ many ratios and then in Line 20 (sparsification step) we crunch them into $m$ ratios. Finally, and similarly to $S_{\text{stat}}(R_i)$, the output $S_{\text{stat}}\left(\widetilde{Y}_n\right)$ can be computed in time $O(m)$ by utilizing the fact that $\widetilde{Y}_n$ may assume $m+1$ values (as there are $m+1$ intervals in $J$).

Therefore, the running time of Algorithm 1 is

$$O(n\ell m) = O(n\ell \left(\log(B/\gamma)\right) / \log(1+\delta))$$
$$= O\Big(n\ell \left(\log\big((1/\tau^n) / \big(\tau^{2n} \left(\varepsilon/2\right) \delta / \left(n \left(1+\delta\right)^n\right)\big)\big)\right) / \log\Big(1 + (1+\varepsilon/2)^{1/n} - 1\Big)\Big)$$
$$= \widetilde{O}\big(n^3\ell \log((1+\varepsilon) / (\varepsilon\tau)) / \varepsilon\big).$$

**Correctness.** We will prove the this algorithm outputs a quantity that is an $\varepsilon$-approximation to $S_{\text{stat}}(P, Q)$. This is accomplished by Lemma 9 and Lemma 10.

**Lemma 9.** *We have that* $S_{\text{stat}}\left(\widetilde{Y}_n\right) \leq (1+\varepsilon) S_{\text{stat}}(P, Q)$.

*Proof.* We will first show that $S_{\text{stat}}\left(\widetilde{Y}_i \cdot Z_i\right) \leq (1+\delta) S_{\text{stat}}(Y_i \cdot Z_i) + \gamma B$. To this end, we have

$$S_{\text{stat}}\left(\widetilde{Y}_i \cdot Z_i\right) = \mathbf{E}\left[\min\left(\widetilde{Y}_i \cdot Z_i, 1\right)\right]$$
$$= \mathbf{E}\left[\sum_{j=0}^{m} \min\left(\widetilde{Y}_i \cdot Z_i, 1\right) \mathbb{1}[Y_i \in I_j]\right]$$
$$= \mathbf{E}\left[\sum_{j=1}^{m} \min\left(\widetilde{Y}_i \cdot Z_i, 1\right) \mathbb{1}[Y_i \in I_j]\right] + \mathbf{E}\left[\min\left(\widetilde{Y}_i \cdot Z_i, 1\right) \mathbb{1}[Y_i \in I_0]\right]$$
$$\leq \sum_{j=1}^{m} \mathbf{E}\left[\min\left(\widetilde{Y}_i \cdot Z_i, 1\right) \mathbb{1}[Y_i \in I_j]\right] + \gamma B$$
$$\leq \sum_{j=1}^{m} \mathbf{E}[((1+\delta) \min(Y_i \cdot Z_i, 1)) \mathbb{1}[Y_i \in I_j]] + \gamma B$$
$$\leq \sum_{j=1}^{m} (1+\delta) \mathbf{E}[\min(Y_i \cdot Z_i, 1) \mathbb{1}[Y_i \in I_j]] + \gamma B$$
$$= (1+\delta) \sum_{j=1}^{m} \mathbf{E}[\min(Y_i \cdot Z_i, 1) \mathbb{1}[Y_i \in I_j]] + \gamma B$$
$$\leq (1+\delta) \sum_{j=0}^{m} \mathbf{E}[\min(Y_i \cdot Z_i, 1) \mathbb{1}[Y_i \in I_j]] + \gamma B$$
$$= (1+\delta) \mathbf{E}[\min(Y_i \cdot Z_i, 1)] + \gamma B = (1+\delta) S_{\text{stat}}(Y_i \cdot Z_i) + \gamma B,$$

The first part of the first inequality follows from the linearity of the expectation. For the second part, note that since $I_0 = (0, \gamma]$, the maximum value $\widetilde{Y}_i$ can take in $I_0$ is at most $\gamma$. The maximum value of $Z_i$ is $B$, thus $\mathbf{E}\left[\min\left(\widetilde{Y}_i \cdot Z_i, 1\right) \mathbb{1}[Y_i \in I_0]\right] \leq \gamma B$. For the second inequality, suppose that $Y_i \in I_j = (\gamma(1+\delta)^{j-2}, \gamma(1+\delta)^{j-1}]$. The maximum value $\widetilde{Y}_i$ can take is $\gamma(1+\delta)^{j-1}$ and the $Y_i$ is larger than $\gamma(1+\delta)^{j-2}$. Thus $\widetilde{Y}_i \leq (1+\delta) Y_i$.

Therefore,

$$S_{\text{stat}}(\widetilde{Y}_n) = S_{\text{stat}}(\widetilde{Y}_n \cdot Z_n) \leq \gamma B + (1+\delta)S_{\text{stat}}(Y_n \cdot Z_n) = \gamma B + (1+\delta)S_{\text{stat}}(\widetilde{Y}_{n-1} \cdot Z_{n-1})$$

so that inductively, we will get

$$S_{\text{stat}}(\widetilde{Y}_n) \leq \gamma B \sum_{k=0}^{n-2} (1+\delta)^k + (1+\delta)^{n-1} S_{\text{stat}}(Y_1 \cdot Z_1)$$

$$\leq \gamma B (1+\delta)^n / \delta + (1+\delta)^n S_{\text{stat}}(P, Q),$$

since $S_{\text{stat}}(Y_1 \cdot Z_1) = S_{\text{stat}}(R_1 \cdot \ldots \cdot R_n) = S_{\text{stat}}(P, Q)$. What is left is to show that $\gamma B (1+\delta)^n / \delta + (1+\delta)^n S_{\text{stat}}(P, Q) \leq (1+\varepsilon) S_{\text{stat}}(P, Q)$. However, this readily follows from the definitions of $\gamma, \delta, B$ as well as Proposition 3 and Proposition 4. Let us elaborate on these calculations. By the fact that $\delta = (1+\varepsilon/2)^{1/n} - 1$, we get $(1+\delta)^n S_{\text{stat}}(P, Q) \leq (1+\varepsilon/2) S_{\text{stat}}(P, Q)$. So what is left is to show that $\gamma B (1+\delta)^n / \delta \leq (\varepsilon/2) S_{\text{stat}}(P, Q)$. By Proposition 3 and Proposition 4, it would suffice to show that $\gamma B (1+\delta)^n / \delta \leq (\varepsilon/2) \tau^n$, which follows directly from the definitions of $\gamma$ and $B$. $\qquad\square$

Similarly, we have the following.

**Lemma 10.** *We have that* $S_{\text{stat}}\left(\widetilde{Y}_n\right) \geq S_{\text{stat}}(P, Q) / (1+\varepsilon)$.

To prove Lemma 10 we will utilize the following claim (proved in Appendix C).

**Claim 11.** *It is the case that* $S_{\text{stat}}(Y_i \cdot Z_i) \leq (1+\delta) S_{\text{stat}}\left(\widetilde{Y}_i \cdot Z_i\right) + \gamma B$.

*Proof of Lemma 10.* By Claim 11, and since $S_{\text{stat}}\left(\widetilde{Y}_n\right) = S_{\text{stat}}\left(\widetilde{Y}_n \cdot Z_n\right)$, we have

$$S_{\text{stat}}\left(\widetilde{Y}_n\right) \geq S_{\text{stat}}(Y_n \cdot Z_n) / (1+\delta) - \gamma B / (1+\delta).$$

Inductively, we get

$$S_{\text{stat}}\left(\widetilde{Y}_n\right) \geq S_{\text{stat}}(Y_1 \cdot Z_1) / (1+\delta)^n - \gamma B \sum_{k=1}^{n-1} 1 / (1+\delta)^k$$

$$\geq S_{\text{stat}}(P, Q) / (1+\delta)^n - n\gamma B.$$

What is left is to show that $S_{\text{stat}}(P, Q) / (1+\delta)^n - n\gamma B \geq S_{\text{stat}}(P, Q) / (1+\varepsilon)$ which is equivalent to $S_{\text{stat}}(P, Q) (1+\varepsilon) \geq n\gamma B (1+\delta)^n (1+\varepsilon) + S_{\text{stat}}(P, Q) (1+\delta)^n$. However, similarly to what we did in Lemma 9, this inequality readily follows from the definitions of $\gamma, \delta, B$ as well as Proposition 3 and Proposition 4. $\qquad\square$

This concludes the proof of Theorem 6.

*Remark* 12. We note that the above estimation algorithm also works for estimating similarity for sub-product distributions. Let $P'$ and $Q'$ be product distributions and $P = \alpha P'$ and $Q = \beta Q'$ for constants $\alpha, \beta$. In this case, to estimate similarity between $P$ and $Q$, we will estimate $\mathbf{E}_{x \sim Q'}[\min(\alpha P'(x) / Q'(x), \beta)]$.

*Remark* 13. While our algorithmic technique is inspired by the work of Feng et al. (2024), the details of our algorithm are different. Specifically, the sparsification procedure is defined differently there, tailored to $d_{\text{TV}}$ estimation instead of $S_{\text{stat}}$. The analysis here is more direct and arguably simpler.

## 6 NP-HARDNESS OF ESTIMATING STATISTICAL SIMILARITY

We show that it is NP-hard to efficiently multiplicatively estimate $S_{\text{stat}}(P, Q)$ for arbitrary Bayes net distributions $P, Q$. That is, we prove Theorem 7. We first formally define Bayes nets.

### 6.1 BAYES NETS

For a directed acyclic graph (DAG) $G$ and a node $v$ in $G$, let $\Pi(v)$ denote the set of parents of $v$.

**Definition 14** (Bayes nets). A *Bayes net* is specified by a DAG over a vertex set $[n]$ and a collection of probability distributions over symbols in $[\ell]$, as follows. Each vertex $i$ is associated with a random variable $X_i$ whose range is $[\ell]$. Each node $i$ of $G$ has a Conditional Probability Table (CPT) that describes the following: For every $x \in [\ell]$ and every $y \in [\ell]^k$, where $k$ is the size of $\Pi(i)$, the CPT has the value of $\mathbf{Pr}[X_i = x | X_{\Pi(i)} = y]$ stored. Given such a Bayes net, its associated probability distribution $P$ is given by the following: For all $x \in [\ell]^n$, $P(x)$ is equal to

$$\mathbf{Pr}_P[X = x] = \prod_{i=1}^{n} \mathbf{Pr}_P\left[X_i = x_i | X_{\Pi(i)} = x_{\Pi(i)}\right].$$

Here, $X$ is the joint distribution $(X_1, \ldots, X_n)$ and $x_{\Pi(i)}$ is the projection of $x$ to the indices in $\Pi(i)$.

Note that $P(x)$ can be computed in linear time by using the CPTs of $P$ to retrieve each $\mathbf{Pr}_P\left[X_i = x_i | X_{\Pi(i)} = x_{\Pi(i)}\right]$.

### 6.2 PROOF OF THEOREM 7

The proof is similar to the proof of hardness of approximating TV distance between Bayes net distributions presented in Bhattacharyya et al. (2023). However, the present proof gives more tight relationship between the number of satisfying assignments of a CNF formula and statistical similarity of Bayes net distributions.

The reduction takes a CNF formula $\phi$ on $n$ variables and produces two Bayes net distributions $P$ and $Q$ (with in-degree at most 2) so that

$$S_{\text{stat}}(P, Q) = \frac{|\text{Sol}(\phi)|}{2^n},$$

where $\text{Sol}(\phi)$ is the set of satisfying assignments for $\phi$.

Let $\phi$ be a CNF formula. Without loss of generality, view $\phi$ as a Boolean circuit (with AND, OR, NOT gates) of fan-in at most two with $n$ input variables $\mathcal{X} = \{X_1, \ldots, X_n\}$ and $m$ internal gates $\mathcal{Y} = \{Y_1, \ldots, Y_m\}$. Let $G_\phi$ be the DAG representing this circuit with vertex set $\mathcal{X} \cup \mathcal{Y}$. So in total there are $n + m$ nodes in $G_\phi$. Assume $\mathcal{X} \cup \mathcal{Y}$ is topologically sorted in the order $X_1, \ldots, X_n, Y_1, \ldots, Y_m$, whereby $Y_m$ is the output gate. For every internal gate node $Y_i$, there is directed edge from node $Y_i$ to node $Y_j$ if the gate/variable corresponding to $Y_i$ is an input to $Y_j$.

We will define two Bayes net distributions $P$ and $Q$ on the same DAG $G_\phi$. Let $X_i$ be a binary random variable corresponding to the input variable node $X_i$ for $1 \le i \le n$ and $Y_j$ be a binary random variable corresponding to the internal gate $Y_j$ for $1 \le j \le m$. The distributions $P$ and $Q$ on $G$ are given by Conditional Probability Tables (CPTs) defined as follows. The CPTs of $P$ and $Q$ will only differ in $Y_m$.

For both $P$ and $Q$, each $X_i$ ($1 \le i \le n$) is a uniform random bit. For each $Y_i$ ($1 \le i \le m-1$), its CPT is the deterministic function defined by its associated gate. For example, if $Y_i$ is an OR gate in $G_\phi$, then $Y_i = 1$ with probability 1 except when the inputs are 00, in which case $Y_i = 0$ with probability 1. The CPTs for AND and NOT nodes are similar.

For $P$, the value of $Y_m$ is given by the deterministic function of the output gate $Y_m$ in $G_\phi$. For $Q$, the value of $Y_m$ is 1 (independently of the input).

Note that even though the sample space is $\{0,1\}^{n+m}$, there are only $2^n$ strings in the support of $P$ and $Q$. In particular, a point $z$ in the sample space $\{0,1\}^{n+m}$ can be written as $xy$ where $x$ is the first $n$ bits and $y$ is the last $m$ bits. By construction, it is clear that for every $x$, there is only one $y$ (which are the gate values for the input assignment $x$) for which $xy$ has positive probability in both the distributions, and this probability is exactly $\frac{1}{2^n}$. For any $x$, let $f_P(x)$ (respectively, $f_Q(x)$) denote this unique $y$ in $P$ (respectively, in $Q$). The crucial observation is that $f_P(x)$ and $f_Q(x)$ are the same if and only if $x$ is in $\text{Sol}(\phi)$. In this case, denote $f_P(x) = f_Q(x) = f(x)$.

Consider $z = xy \in \{0,1\}^{n+m}$. If $xy$ is not in the support of both $P$ and $Q$, then the minimum of $P(z)$ and $Q(z)$ is 0, so assume that $xy$ is in the support of at least one.

**Case 1.** Assume that $x$ is a not a satisfying assignment of $\phi$. Then $z = x f_P(x)$ is in the support of $P$, however, it is not in the support of $Q$ as the last bit of $z = 0$. Similarly $x f_Q(x)$ is in the support of $Q$ but not in $P$. Hence, $\min(P(z), Q(z)) = 0$.

**Case 2.** Assume that $x$ is a satisfying assignment of $\phi$. In this case the last bit of $z = x f_P(x)$ is 1 and hence is in the support on both $P$ and $Q$ and has a probability of $\frac{1}{2^n}$. Thus $\min(P(z), Q(z)) = \frac{1}{2^n}$. Hence we have

$$
S_{\text{stat}}(P, Q) = \sum_{z \in \{0,1\}^{n+m}} \min\left(P(z), Q(z)\right)
$$

$$
= \sum_{x \in \text{Sol}(\phi)} \min(P(xf(x)), Q(xf(x)) + \sum_{x \notin \text{Sol}(\phi)} \min(P(xf_P(x)), Q(xf_P(x)))
$$

$$
+ \sum_{x \notin \text{Sol}(\phi)} \min(P(xf_Q(x)), Q(xf_Q(x))) = \frac{|\text{Sol}(\phi)|}{2^n} + 0 + 0 = \frac{|\text{Sol}(\phi)|}{2^n}.
$$

This concludes the proof.

## 7 CONCLUSION

Statistical similarity ($S_{\text{stat}}$) between distributions is a fundamental quantity. In this work, we initiated a computational study of $S_{\text{stat}}$. Prior results on statistical distance computation imply that the exact computation of $S_{\text{stat}}$ for high-dimensional distributions is computationally intractable.

Our first contribution is a fully polynomial-time deterministic approximation scheme (FPTAS) for estimating statistical similarity between two product distributions. Notably, the existing FPTAS for statistical distance (Feng et al., 2024) does not directly yield an FPTAS for $S_{\text{stat}}$. We also establish a complementary hardness result: Approximating $S_{\text{stat}}$ for Bayes net distributions is NP-hard. Extending our results beyond product distributions to more structured settings, such as tree distributions, remains a significant and promising research direction.

We believe $S_{\text{stat}}$ computation is a compelling problem from a complexity theory perspective. Interestingly, for product distributions, both $S_{\text{stat}}$ and its complement ($1 - S_{\text{stat}} = d_{\text{TV}}$) admit FPTAS, making it one of the rare problems with this property. A deeper complexity-theoretic study of functions $f$ in #P with range in $[0, 1]$, where both $f$ and $1 - f$ have approximation schemes, is an intriguing direction for future research. Finally, this work is limited to the algorithmic foundational aspects of $S_{\text{stat}}$ computation. We leave the experimental evaluation of the algorithm for future work.

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

## A  BAYES ERROR AND STATISTICAL SIMILARITY

A *binary prediction problem* is a distribution $P$ of $X \times \{0, 1\}$ where $X$ is a (finite) feature space. A *classifier* is a deterministic function $g : X \to \{0, 1\}$. The 0-1 *error of the predictor* $g$ is $\mathbf{Pr}_{(x,y)\sim P}[g(x) \neq y]$.

The *Bayes optimal classifier* is the classifier that outputs 1 if and only if $P(1|x) \geq P(0|x)$. The error of Bayes optimal classifier denoted as $R^*$ is called the *Bayes error*. Bayes error is the minimum

error possible in the sense that the error of any classifier is at least Bayes error. It is known that for a prediction problem $P$, its Bayes error $R^*$ is given by the following marginal expectation:

$$R^* = \mathbf{E}_X[\min(P(0|X), P(1|X))].$$

Let the prior probabilities $P(0)$ and $P(1)$ be denoted by $\alpha_0$ and $\alpha_1$ respectively. Note that $\alpha_0$ and $\alpha_1$ are constants that sum up to 1. We call $P$ *balanced* if $P(0) = P(1) = 1/2$. For simplifying notation, we will also denote the likelihood distributions $P(X|0)$ and $P(X|1)$ as $P_0$ and $P_1$ respectively.

**Theorem 15.** *For a prediction problem $P$, its Bayes error is given by*

$$R^* = S_{\text{stat}}(\alpha_0 P_0, \alpha_1 P_1).$$

*In particular, for a balanced prediction problem $P$, its Bayes error is given by $R^* = S_{\text{stat}}(P_0, P_1)/2$.*

*Proof.* The proof is a simple application of the Bayes theorem. That is,

$$\begin{aligned}
R^* &= \mathbf{E}_X[\min(P(0|X), P(1|X)] \\
&= \sum_{x \in X} P(x) \cdot \min(P(0|x), P(1|x)) \\
&= \sum_x P(x) \cdot \min\left(\frac{P(x|0)P(0)}{P(x)}, \frac{P(x|1)P(1)}{P(x)}\right) \\
&= \sum_x \min(P(x|0)P(0), P(x|1)P(1)) = S_{\text{stat}}(\alpha_0 P_0, \alpha_1 P_1).
\end{aligned}$$

The balanced case follows from the fact that $\alpha_0 = \alpha_1 = \frac{1}{2}$. $\qquad\square$

## B  TOTAL VARIATION DISTANCE AND STATISTICAL SIMILARITY

Statistical similarity can be proved to be equal to the complement of statistical distance, which is commonly called total variation distance (denoted by $d_{\text{TV}}$). See below.

**Definition 16.** For distributions $P, Q$ over a sample space $D$, the *total variation (TV) distance between $P$ and $Q$* is

$$d_{\text{TV}}(P, Q) := \sum_{x \in D} \max(0, P(x) - Q(x)).$$

**Proposition 17** (Scheffé's identity, see also (Tsybakov, 2009)). *Let $P, Q$ be distributions over a sample space $D$. Then $S_{\text{stat}}(P, Q) = 1 - d_{\text{TV}}(P, Q)$.*

*Proof.* We have that

$$\begin{aligned}
S_{\text{stat}}(P, Q) &= \sum_{x \in D} \min(P(x), Q(x)) \\
&= \sum_{x \in D} \min(P(x), P(x) + Q(x) - P(x)) \\
&= \sum_{x \in D} P(x) + \sum_{x \in D} \min(0, Q(x) - P(x)) \\
&= 1 - \sum_{x \in D} \max(0, P(x) - Q(x)) = 1 - d_{\text{TV}}(P, Q). \qquad\square
\end{aligned}$$

## C  PROOF OF CLAIM 11

We have

$$S_{\text{stat}}(Y_i \cdot Z_i) = \mathbf{E}[\min(Y_i \cdot Z_i, 1)]$$

$$= \mathbf{E}\left[\sum_{j=0}^{m} \min(Y_i \cdot Z_i, 1)\, \mathbb{1}[Y_i \in I_j]\right]$$

$$= \mathbf{E}\left[\sum_{j=1}^{m} \min(Y_i \cdot Z_i, 1)\, \mathbb{1}[Y_i \in I_j]\right] + \mathbf{E}[\min(Y_i \cdot Z_i, 1)\, \mathbb{1}[Y_i \in I_0]]$$

$$\leq \sum_{j=1}^{m} \mathbf{E}[\min(Y_i \cdot Z_i, 1)\, \mathbb{1}[Y_i \in I_j]] + \gamma B$$

$$\leq \sum_{j=1}^{m} \mathbf{E}\left[\left((1+\delta)\min\left(\widetilde{Y}_i \cdot Z_i, 1\right)\right) \mathbb{1}[Y_i \in I_j]\right] + \gamma B$$

$$\leq \sum_{j=1}^{m} (1+\delta)\, \mathbf{E}\left[\min\left(\widetilde{Y}_i \cdot Z_i, 1\right) \mathbb{1}[Y_i \in I_j]\right] + \gamma B$$

$$= (1+\delta)\sum_{j=1}^{m} \mathbf{E}\left[\min\left(\widetilde{Y}_i \cdot Z_i, 1\right) \mathbb{1}[Y_i \in I_j]\right] + \gamma B$$

$$\leq (1+\delta)\sum_{j=0}^{m} \mathbf{E}\left[\min\left(\widetilde{Y}_i \cdot Z_i, 1\right) \mathbb{1}[Y_i \in I_j]\right] + \gamma B$$

$$= (1+\delta)\, \mathbf{E}\left[\min\left(\widetilde{Y}_i \cdot Z_i, 1\right)\right] + \gamma B = (1+\delta)\, S_{\text{stat}}\left(\widetilde{Y}_i \cdot Z_i\right) + \gamma B,$$

The first part of the first inequality follows from the linearity of the expectation. For the second part, note that the interval $I_0 = (0, \gamma]$. Thus, the maximum value $Y_i$ can take in this interval is almost $\gamma$. The maximum value of $Z_i$ is $B$, thus $\mathbf{E}[\min(Y_i \cdot Z_i, 1)\, \mathbb{1}[Y_i \in I_0]] \leq \gamma B$. For the second inequality, suppose that $Y_i \in I_j = (\gamma(1+\delta)^{j-2}, \gamma(1+\delta)^{j-1}]$. The maximum value $Y_i$ can take is $\gamma(1+\delta)^{j-1}$ and the $\widetilde{Y}_i$ is larger than $\gamma(1+\delta)^{j-2}$. Thus $Y_i \leq (1+\delta)\, \widetilde{Y}_i$.

