# OpenReview forum: "Algorithms and Hardness for Estimating Statistical Similarity"
_ICLR.cc/2026/Conference — Submitted to ICLR 2026_

### Official Review · Reviewer_Ji8F · 2025-10-30

**Soundness:** 3
**Presentation:** 3
**Contribution:** 2
**Rating:** 4
**Confidence:** 4

**Summary:**

This paper studies the problem of estimating the statistical similarity between two discrete probability distributions, defined as the sum (over all points in the support) of the smaller of the two probabilities at each point. This quantity is closely related to the well-known total variation (TV) distance, since the two are exact complements: knowing one determines the other. While additive approximations of TV distance translate directly to additive approximations of statistical similarity, this correspondence does not extend to relative (multiplicative) approximations.

The paper focuses on computing statistical similarity between product distributions over the corners of a unit hypercube, i.e., $[0,1]^d$, where each coordinate has an independent one-dimensional marginal. Since the exact computation of TV distance between such product distributions is known to be #P-hard, the same hardness applies to statistical similarity. The authors present a fully polynomial-time approximation scheme (FPTAS) for estimating statistical similarity in this setting. An FPTAS for the TV distance between product distributions was previously known, and my understanding is that the proposed algorithm extends and adapts those ideas to handle the statistical similarity formulation.

The paper also establishes that obtaining a relative approximation is NP-hard for Bayesian networks of in-degree two, complementing the positive result by showing that multiplicative approximations are generally intractable. Together, these results delineate the boundary between tractable and intractable cases for this class of problems.

My main concern is the level of algorithmic novelty relative to prior work on TV distance approximation—the techniques appear to be reparameterization of the TV-distance approximation rather than containing any fundamentally new algorithmic ideas. Nevertheless, the results are technically sound and provide a clear and useful extension, showing that statistical similarity can indeed be approximated efficiently in polynomial time.

**Strengths:**

The results are technically sound and provide a clear and useful extension of TV-distance, showing that statistical similarity can indeed be approximated efficiently in polynomial time.

**Weaknesses:**

The level of algorithmic novelty relative to prior work on TV distance approximation is small—the techniques appear to be reparameterization of the TV-distance approximation rather than containing any fundamentally new algorithmic ideas

**Questions:**

NA

---

> ### Author Response · Authors · 2025-11-26
>
> Dear Reviewer,
>
> Thank you very much for your feedback.
>
> While our method relies on the usage of ratio distributions, which was also used for TV distance approximation. Most of the FPTAS known have to argue about some sort of sparsification but these similarities are rather superficial since the ratio distributions for TV distance are fundamentally different objects compared to that of statistical similarity. In particular, none of the technical lemmas follow from the work on TV distance approximation.

---

### Official Review · Reviewer_xZMx · 2025-10-31

**Soundness:** 2
**Presentation:** 3
**Contribution:** 3
**Rating:** 4
**Confidence:** 2

**Summary:**

The paper examines the computational complexity of approximating the statistical similarity of two distributions $P,Q$ on a finite sample space. This quantity can be defined as $S_{\text{stat}}(P,Q) = 1-d_{\text{TV}}(P,Q)$, where $d_{\text{TV}}$ is the total variation distance. Previous work has shown that an FPTAS exists for the TV-distance between product distributions. However, proving this existence for statistical similarity is not straightforward, as the existence of an FPTAS for a function $f$ need not imply the same for $1-f$. The paper resolves this question of existence in the affirmative by constructing an explicit FPTAS for $S_{\text{stat}}(P,Q)$ when $P,Q$ are product distributions. The authors further demonstrate that this property is ``sharp'' in the sense that computing a multiplicative approximation for a slightly more complex family of distributions (Bayes nets of in-degree two) is NP-hard.

**Strengths:**

The paper defines an FPTAS for computing the statistical similarity between product distributions, and confirms the NP-hardness of obtaining a multiplicative approximation for distributions given by Bayes nets of in-degree two. The work fills an interesting theoretical gap, complementing similar results for the total variation distance. The description of the algorithm, and the proofs of the various results are clear and easy to follow.

**Weaknesses:**

There are some potential issues with the proof of Theorem 6 (existence of an FPTAS algorithm) that I have highlighted below. I would be willing to provide a more positive score if these can be resolved. Perhaps some comments on the practical aspects of the proposed algorithm relative to the FPTAS for the TV distance (or other estimation techniques for this) could further motivate the work.

**Questions:**

Concerning the proof of Theorem 6 (existence of an FPTAS):

1. In the proof of Lemma 9, moving from line 327 to 329 requires the-inequality $\sum^{n-2}_{k=0}(1+\delta)^k \leq (1+\delta)^n$ where $\delta = \epsilon/2n$. I believe one can show that for a fixed $\epsilon$, the LHS is unbounded in $n$ while the RHS is bounded.

2. In the same proof, moving from line 329 to 330 requires $(1+\delta)^n \leq (1+n \delta)$. This is contradicted by Bernoulli's inequality (for $n>1,\delta>0$, asserts $>$ holds rather than $\leq$).

3. The same thing occurs in lines 335-336 where it is claimed that $(1+\delta)^n S_{\text{stat}}(P,Q) \leq (1+\epsilon/2) S_{\text{stat}}(P,Q)$.

4. I would appreciate some clarification on how the inequality $S_{\text{stat}}(P,Q)(1+\epsilon) \geq 2n^2 \gamma B (1+\epsilon) + S_{\text{stat}}(P,Q)(1+\delta)^n$ on lines 361-362 is attained. The text claims that this uses similar reasoning to Lemma 9, which may be erroneous given the above.

Minor issues (presentation, typos, etc.):

5. I would appreciate some further detail on how the final expression for the computational complexity of the FPTAS algorithm is obtained (lines 278-282). Perhaps this could be included in the supplementary material.

6. In the proof of Lemma 9, the chain of (in)equalities from lines 289 to 315 could be condensed without affecting readability. For examples, lines 294 and 297 could be combined, as could 306 to 312.

7. In lines 306 and 607, moving the constant through the ``min'' function should lead to a $\leq$ sign rather than an equality.

---

> ### Author Response · Authors · 2025-11-21
>
> Dear Reviewer,
>
> Thank you very much for your feedback.
>
> We have amended the issues that you pointed out, regarding some of the inequalities that appear in our work.
> Our main changes are as follows:
>
> 1. We are now bounding $\sum^{n - 2}_{k = 0}(1 + \delta)^k$ by $(1 + \delta)^n / \delta$, by appealing to the partial sum formula of a geometric series.
>
> 2. We are no longer using the (erroneous) inequality $(1 + \delta)^n \leq (1 + n \delta)$.
>
> 3. We have now set $\delta := (1 + \varepsilon / 2)^{1 / n} - 1$ so that $(1 + \delta)^n \leq (1 + \varepsilon / 2)$.
>
> Please do let us know if there is anything that is still unclear. Thank you :)

---

### Official Review · Reviewer_8TVK · 2025-11-01

**Soundness:** 4
**Presentation:** 3
**Contribution:** 2
**Rating:** 6
**Confidence:** 3

**Summary:**

This paper studies the problem of computing the statistical similarity between two distributions. This is defined as S(P,Q) = sum_{x in support} min(P(x), Q(x)). This natural notion can be interpreted as the error achieved by an optimal Bayes classified (for the similarity of the label 0 and label 1 distributions) and similarly in hypothesis testing.

The paper studies two new results about the computational complexity.

First, for product distributions, it gives the first FPTAS. (Note that the product distribution case is specifically the one which applies to hypothesis testing.) The problem was previously known to be #P hard to compute exactly in this case.

Second, if P and Q come from Bayes nets, then it shows it is NP-hard to estimate the similarity.

**Strengths:**

The problems studied here are fundamental. The paper provides strong theoretical results with clean proofs, both for algorithms and for lower bounds. The introduction motivated well that these are important problems to study, particularly these special cawses (product distribution adn Bayes net) which I was skeptical of at first.

**Weaknesses:**

My main concern is that the proof techniques apper quite standard or follow from prior work. Could you comment on where the technical novelty is? In particular, why does the result of Bhattacharyya et al. (2023) not already essentially imply Theorem 7?

**Questions:**

Can we extend the results to the case when Q is known and only P is unknown? Such as testing whether a Bayes net has desired behavior?

---

> ### Author Response · Authors · 2025-11-23
>
> Dear Reviewer,
>
> Thank you very much for your feedback.
>
> To address you main concern, please note that we are including the following passage in our paper, which explains why the result of Bhattacharyya et al. (2023) does not imply our Theorem 7:
>
> >It is perhaps worth remarking that technical barrier in translating multiplicative approximation of statistical distance to statistical similarity is rather fundamental, i.e., it is not possible in general to use an efficient multiplicative approximation algorithm for a function $f$ in order to design an efficient multiplicative approximation algorithm for $1 - f$.
> In particular, even if there is an efficient multiplicative approximation algorithm $f$, approximating $1 - f$ could be $\mathsf{NP}$-hard.
> For instance, let $f$ be a function that takes as input a Boolean DNF formula $\phi$ and outputs the probability that a random assignment satisfies $\phi$.
> It is known that there is a randomized multiplicative approximation algorithm for estimating $f$ [1].
> However, a multiplicative approximation algorithm for estimating $1 - f$ implies that all $\mathsf{NP}$-complete problems have efficient randomized algorithms ($\mathsf{RP} = \mathsf{NP}$).
> This is because the complement of a DNF formula is a CNF formula, and there is no efficient randomized multiplicative approximation for estimating the acceptance probability of CNF formulas unless $\mathsf{RP} = \mathsf{NP}$.
>
> We acknowledge your other question about extending our results to the case when $Q$ is known and $P$ is unknown, as putting forward an interesting open problem :)
>
> [1] Richard M. Karp, Michael Luby, and Neal Madras. Monte-Carlo approximation algorithms for enumeration problems. J. Algorithms, 10(3):429–448, 1989.

---

### Official Review · Reviewer_kgs5 · 2025-11-01

**Soundness:** 3
**Presentation:** 2
**Contribution:** 1
**Rating:** 2
**Confidence:** 3

**Summary:**

This paper solves the following problem: There are two distributions $P$ and $Q$ which are product distributions over $[\ell]^n$. The goal is to "estimate" $\sum_x \min(P(x),Q(x))$, where the sum runs over the entire domain. Given any fixed precision parameter $\epsilon \in (0,1)$, the paper proposes an algorithm to compute a $1+\epsilon$ multiplicative approximation of this quantity. The algorithm assumes query access to the underlying probability values of all the marginals and it is not clear what the exact query complexity dependence is in the main Theorem 6.

**Strengths:**

Estimation properties of distributions is a fundamental topic in machine learning and at least the "complement" problem of estimating the TV distance is a well studied topic.

**Weaknesses:**

A major weakness of the paper is that it feels slightly misleading for the following reasons: There is actually nothing 'statistical' about the paper as the title/intro suggests. The algorithm is not sampling from the distributions! Rather, this is a purely query complexity result where the result assumes knowledge of the underlying probability values. There is no notion of sample complexity which is what the title 'statistical similarity' would suggest.

In fact, this problem does not seem to admit any meaningful approximation using samples. This is perhaps why the authors are assuming access to the probability values? But this should be spelled out in the text as getting samples is usually the standard way of accessing distributions. The reason this problem is difficult with samples is that the quantity $\sum_x \min(P(x),Q(x))$ can be very close to 0 (e.g. imagine distributions that are supported on disjoint subsets) and we would not figure this out unless we took many samples. Indeed, as the authors point out, this relates to estimating the total variation distance which is known to require samples almost equal to the domain size.

But I don't see why assuming the knowledge of the exact marginal probabilities (in the case that P and Q are product distributions) is a natural assumption. In fact, I also don't see why assuming P and Q are product distributions in the first place is a natural assumption. This makes the problem feel very niche (note that the interesting cases of TV estimation assumes only sample access and it is known that one needs sample complexity scaling with the domain size to obtain additive error).

Thus it is unclear to me if this is an interesting result for the learning theory community since there is no aspect of sample complexity.

The presentation of the main theorem statement also leaves a bit to be desired. Theorem 6 is written extremely informally and does not state what the exact epsilon dependence is on the query complexity (of how many of the underlying probabilities are queried). It also seems to 'hide' the PDF access model by saying "product distributions succinctly represented by their component distributions" , which should not be appropriate for a formal theorem statement.

**Questions:**

See above

---

> ### Author Response · Authors · 2025-11-19
> **Unacceptable Review**
>
> Dear Reviewer,
>
> We would like to humbly but strongly disagree with you, and we are even of the opinion that reviews such as this have the potential to ultimately kill any trust in the peer review system.
>
> We have been in this rodeo for a long time and therefore have experienced many paper rejections, and naturally one is often disappointed when one receives a negative review. But your review stands out.
>
> The review begins with: "There is actually nothing 'statistical' about the paper as the title/intro suggests. The algorithm is not sampling from the distributions." Let's revisit the title: "Algorithms and Hardness for Estimating Statistical Similarity." A title that is "Algorithms and Hardness for X" will focus on the problem X and provide algorithm and hardness results for that problem. In this paper, we focus on a problem called statistical similarity, which is defined in the second line of the introduction.
>
> Saying that you saw the word "statistical" and expected sampling is like saying that a paper on "treewidth" should talk about "trees" in natural forests and discuss how to measure the width of the trunks of those trees. It is perfectly understandable for a reviewer not to know a well-known term in the title, but it is certainly unacceptable to demand that the term's meaning be defined as what the reviewer demands rather than what the rest of the community defines it as.
>
> The review then mentions: "But I don't see why assuming the knowledge of the exact marginal probabilities (in the case that P and Q are product distributions) is a natural assumption... Thus it is unclear to me if this is an interesting result for the learning theory community since there is no aspect of sample complexity." Except if you had looked at the learning theory community's papers, there are many papers that focus on the case where the parameters of the distributions are explicitly specified and we are interested in questions about these distributions.
>
> In fact, one of the most interesting recent results in the area has been the result that shows #P-hardness of total variation distance for product distributions, which has inspired a lot of follow-up work (see: Feng, Guo, Jerrum, and Wang; Feng, Liu, and Liu and discussion therein).
>
> The ACM code of ethics strictly requires that one should work only in areas of competence [1]. You have marked your confidence score as "3: You are fairly confident in your assessment." However, given the glaring flaws in your review—such as not knowing what "statistical similarity" means or asserting that "the learning theory community only cares about sample complexity"—we would like to object to your assessment of your expertise with respect to the technical content of this paper. Therefore, as per the ACM code of ethics, an appropriate action would be to inform the AC [2].
>
> We apologize for coming out so harshly, but we think, as they say: "if you see something, say something," and reviews such as these are indeed a grave threat to the peer review system, and it's best to call them out.
>
> [1] "2.6 Perform work only in areas of competence: A computing professional is responsible for evaluating potential work assignments. This includes evaluating the work's feasibility and advisability, and making a judgment about whether the work assignment is within the professional's areas of competence." (https://www.acm.org/code-of-ethics)
>
> [2] "If at any time before or during the work assignment the professional identifies a lack of a necessary expertise, they must disclose this to the employer or client. The client or employer may decide to pursue the assignment with the professional after additional time to acquire the necessary competencies, to pursue the assignment with someone else who has the required expertise, or to forgo the assignment." (https://www.acm.org/code-of-ethics)

---

> > ### Comment · Reviewer_kgs5 · 2025-11-20
> > **Follow up to authors**
> >
> > I believe my statement that the submission should not be categorized under algorithmic statistics is quite fair. Rather, the paper falls under a 'query complexity' model. Note that PDF queries have been well studied in the algorithmic statistics literature, but they are usually coupled with sampled access and are used to contrast the power of using only vs using samples + pdf queries. The submission does not require the underlying functions to be distributions because *they are not being accessed as distributions*. Rather, they are functions which decompose as product over coordinates. Thus, I am not convinced that this setting is motivated from a statistical view point for the following reasons:
> >
> > - Product distributions are natural but the paper never addresses the more basic question of why samples are not sufficient for estimating TV distance or statistical similarity for product distributions.
> > - Why is having PDF access to these types of distribution natural? If PDF access is assumed, then a more general result beyond product distributions would be more interesting.
> >
> > Ultimately the goal of the paper is to estimate something quite challenging since it is related to the TV distance. But it is well known that TV distance estimation is hard statistically (the number of samples scales with the domain size) since one needs samples that scale with the domain size in the worst case. However, estimating TV distance is *very easy* if one has sample access + PDF query access (See corollary 2 in https://arxiv.org/pdf/1402.3835). The authors do not seem to be aware of this. Thus the "more" natural questions (upper/lower sample complexity bounds with only samples and upper/lower sample complexity bounds with samples + PDF access) should be studied first before this work.
> >
> >  (Note that the hardness given in the paper does not imply anything about product distributions).
> >
> > I urge the authors to try to update the paper to address actionable items including motivating the problem from a statistical viewpoint, discussing the above more natural regimes, and focusing on the technical aspects of the review above including making the main theorem statement more precise.

---

> ### Author Response · Authors · 2025-11-20
> **Follow-up to Reviewer: Serious Misunderstandings**
>
> Dear Reviewer,
>    We thank you for your response but we must point out that  your assertions are still wrong as we discuss below:
>
> Corollary 2 in https://arxiv.org/pdf/1402.3835 refers to additive approximation. Additive approximation of TV (or statistical similarity) is easy given sample access but that's not the case for relative approximation. See page 4 at https://arxiv.org/pdf/2309.09134 for more discussion.
> Sampling+Eval (pdf evaluation) won't work in the case when the distance is very small, in such a case, we would need O(1/distance) many samples.
>
> The additive approximation of f also provides additive approximation of 1-f and therefore, the questions that you are pointing ("Thus the "more" natural questions (upper/lower sample complexity bounds with only samples and upper/lower sample complexity bounds with samples + PDF access) should be studied first before this work.") are already studied. We also point this out in line 157-160.
>
>
> No one would ever argue against your assertion "If PDF access is assumed, then a more general result beyond product distributions would be more interesting." Of course, it would be more interesting to have the result for arbitrary distributions but we don't have FPTAS for total variation distance (arguably more studied problem) for any distributions beyond product, so product is indeed the correct class to focus on for now.
>
>
> Now that we have again pointed out that your assertions are wrong, we hope you will consider rethinking about the paper. otherwise, it looks like you are arguing against the paper solely based on an expression stated on line 282 not stated on line 190 as that remains the *only* technical aspect of your review which is not technically incorrect.

---

### Meta-Review · Area_Chair_afjc · 2025-12-28

**Summary:**

The initial reviews are not positive. After reading the rebuttal, I would think some reviewer may increase the score towards acceptance, because I find there are major misconceptions clarified in the rebuttal. Nonetheless, there are still issues which I don't think fully resolved. Overall, I find this paper interesting, but there are still some nontrivial issues, which makes me leaning to reject. The more detailed concerns are listed in the following.

**Reviewer Concerns:**

There seems to be some misconceptions from the reviewers, and they are sorted out from the rebuttal:

1. Reviewer kgs5 challenged the sampling complexity, and after reading the rebuttal I think the authors are correct and the reviewer's focus for evaluating the paper is not proper.

2. Reviewer 8TVK asked if the main result can be readily implied by a previous paper which is about statistical distance. This is clearly reflected by the authors that the statistical similarity is (1 - distance) so the ratio is not preserved.

However, I find the following issues outstanding and is not satisfactorily addressed by the authors.

1. The motivation of considering product distribution. While I understand there are solid reasons from theory perspective, but this is ICLR and it is important to also discuss the practical aspects, which seems to be missing.

2. Several technical issues are discovered by Reviewer xZMx. Albeit the mentioned points seem to be addressed by the authors in a follows-up message, it is still possible that the new proof is not completely correct. To verify the correctness, we may need the reviewer's careful reading and response, which is unfortunately not possible. In general, this is a sign of the quality issue of the paper, and I am not confident about the correctness. This makes me hesitant to allow the paper to be published.

3. Ji8F challenged the technical novelty. The response is not informative enough. To me, it looks like they only differ in technical details, and no high-level/fundamental difference is discussed. It is useful to give a specific yet fundamental technical aspect to illustrate the challenge. That being said, the technical novelty is probably not the most important criteria for ICLR.

**Reviewer Scores:**

Reviewer 8TVK may raise the score because this reviewer is supportive and the response clarified an important point that shows the paper is nontrivial. Other reviewers probably do not change their scores, either because the response is not convincing enough, or because they seem to have strong opinion about the paper.

---

### Decision · Program_Chairs · 2026-01-26

Reject